

# DisVar: an R library for identifying variants associated with diseases using large-scale personal genetic information

Khunanon Chanasongkhram[1], Kasikrit Damkliang[2] and Unitsa Sangket[1,3]

[1] Division of Biological Science, Faculty of Science, Prince of Songkla University, Hat Yai, Songkhla, Thailand
[2] Division of Computational Science, Faculty of Science, Prince of Songkla University, Hat Yai, Songkhla, Thailand
[3] Center for Genomics and Bioinformatics Research, Faculty of Science, Prince of Songkla University, Hat Yai, Songkhla, Thailand

## ABSTRACT

**Background**. Genetic variants may potentially play a contributing factor in the development of diseases. Several genetic disease databases are used in medical research and diagnosis but the web applications used to search these databases for disease-associated variants have limitations. The application may not be able to search for large-scale genetic variants, the results of searches may be difficult to interpret and variants mapped from the latest reference genome (GRCH38/hg38) may not be supported.
**Methods**. In this study, we developed a novel R library called "DisVar" to identify disease-associated genetic variants in large-scale individual genomic data. This R library is compatible with variants from the latest reference genome version. DisVar uses five databases of disease-associated variants. Over 100 million variants can be simultaneously searched for specific associated diseases.
**Results**. The package was evaluated using 24 Variant Call Format (VCF) files (215,054 to 11,346,899 sites) from the 1000 Genomes Project. Disease-associated variants were detected in 298,227 hits across all the VCF files, taking a total of 63.58 m to complete. The package was also tested on ClinVar's VCF file (2,120,558 variants), where 20,657 hits associated with diseases were identified with an estimated elapsed time of 45.98 s.
**Conclusions**. DisVar can overcome the limitations of existing tools and is a fast and effective diagnostic and preventive tool that identifies disease-associated variations from large-scale genetic variants against the latest reference genome.

# INTRODUCTION

The genetic information of living things is stored and transmitted by DNA. Differences in DNA sequences between individuals or populations are called variants and may play a role in the development of genetic disorders (*Benton et al., 2021*) that lead to common diseases such as cancer, diabetes, or hypertension. These diseases result from single or combined variations in DNA loci and, given the right environment or lifestyle habits, can promote or increase the severity of a disease (*Jackson et al., 2018*). Variants that are associated with

Corresponding author
Unitsa Sangket, unitsa.s@psu.ac.th

traits in a population are commonly identified in genome-wide association studies (GWAS) (*Tam et al., 2019*). These studies have successfully identified relationships between single nucleotide polymorphisms (SNPs) and disease (*Visscher et al., 2012*). Comparisons with the reference genome have revealed up to 5 million sites where variants exist (*Auton et al., 2015*). Several genetic disease databases provide resources for medical research and diagnosis. Some well-known examples include the GWAS catalogue, GWASdb, the Genome-Wide Repository of Associations Between SNPs and Phenotypes (GRASP), The Genetic Association Database for Centers of Disease Control (GADCDC), and Johnson and O'Donnell's database. The GWAS catalogue summarizes the results of GWASs, looking for correlations between genetic variants and traits or diseases, and contains 71,673 pathogenic variants from 3,567 studies (*Buniello et al., 2019*). GWASdb provides access to summary statistics and individual-level data from GWASs for further research and meta-analysis (*Li et al., 2012*). GRASP provides information on the associations between SNPs and phenotypes, including disease-related traits, and is derived from 2,082 studies containing approximately 8.87 million disease-related variants (*Eicher et al., 2015*). Johnson and O'Donnell's database, based on 118 studies, contains 56,411 variants with high disease specificity, and is used in specific research projects or studies (*Johnson & O'Donnell, 2009*). However, current methods of searching for disease-associated variants have their limitations. For example, the GWAS Catalog web tool is widely used for its convenience and efficiency in searching for single variants but is not suitable for simultaneous searches for multiple variants (*Welter et al., 2014*). Simultaneous searches for multiple variants can be performed with the GWAS4D web application, but it is difficult to interpret the results to identify a disease (*Huang et al., 2018*). The identification of diseases from Variant Call Format (VCF) files is possible with the prediction tool MutationTaster2021, but the program does not support variants from the latest reference genome (GRCh38/hg38) (*Steinhaus et al., 2021*). It is challenging to find all the millions of variants because every web application has a limited capacity for large-scale searches for variants. Moreover, processing takes a long time and variants mapped from the latest reference genome are not supported.

The R language is an open-source, free computational and statistical analysis instrument that is compatible with Windows, Linux, and macOS. It is highly effective at processing large amounts of data, statistical computations, and modelling, and can be easily integrated into pipelines. R packages, developed by a community of programmers, provide a set of functions, datasets and compiled code that simplify complex computations (*Kim et al., 2019*). R packages are stored in libraries and can be selected from a wide range of options (*Sangket et al., 2015*; *Sangket et al., 2022*), such as the sqldf package, which allows SQL statements to be executed on data frames, facilitating database management and variant searching.

In this research, an R package called "DisVar" was created to overcome the limitations of existing web tools in the detection of disease-associated variations from large-scale individual genomic data. This package is compatible with variants from the latest reference genome version and can process large amounts of variants rapidly and effectively. This

**Table 1  An example of a VCF format file (fileformat = VCFv4.1).**

##fileformat=VCFv4.1

| CHROM | POS | ID | REF | ALT |
|---|---|---|---|---|
| 3 | 4700592 | rs6762644 | A | G |
| 6 | 149842694 | rs1125107 | T | C |
| 16 | 5225061 | rs2333967 | C | T |
| X | 83723541 | rs35161124 | A | G |

makes it possible to identify genetic disorders based on individual variations, making disease detection and prevention easier.

## METHODS

### Data preparation

VCF is a text file format used in bioinformatics to store variations in gene sequences input to identify variants associated with diseases. VCF files derived from GRCh38/hg38 were obtained from the 1000 Genomes Project (http://ftp.1000genomes.ebi.ac.uk/vol1/ftp/data_collections/1000G_2504_high_coverage/working/20201028_3202_raw_GT_with_annot) and ClinVar (https://ftp.ncbi.nlm.nih.gov/pub/clinvar/vcf_GRCh38/).

Risk SNP database files were collected from VARAdb, a comprehensive database of human variation annotations that combines the five databases used to develop the DisVar R library: the GWAS catalogue, and the GWASdb, GRASP, GADCDC, and Johnson and O'Donnell's databases (http://www.licpathway.net/VARAdb/download.php).

Table 1 shows an example of a VCF format file. It consists of meta-information lines, a header line, and then data lines, each containing information about variants in the genome. The header line and data lines are separated into multiple columns by tabs. The DisVar library reads data from the CHROM column, which presents the chromosome names, and the POS column, which indicates the position of the variation on that chromosome. The REF column lists the allele found at the POS position in the reference genome, and the ALT column lists the alternate allele in the variation, which differs from the reference allele. The type of variant can be determined by comparing the REF and ALT columns, which can be an SNP, an insertion, a deletion, or a more complex variation. This data can be used to refine and prioritize variants for analysis. The REF and ALT columns, together with the CHROM and POS columns, provide important information for efficiently identifying disease-associated variants, and facilitate the detection of genetic diseases based on the genetic variants of individuals. Since DisVar is compatible with variants from GRCh38/hg38, it can detect the most up-to-date variants.

### Instruments and software

The R package was developed on an ASUS TUF Gaming F15 FX506HCB laptop with a 2.70 GHz Intel Core i5-11400H processor, 32 GB RAM and a 512 GB solid-state drive. The laptop ran Windows 11 Home 64-bit and was equipped with a dedicated NVIDIA GeForce

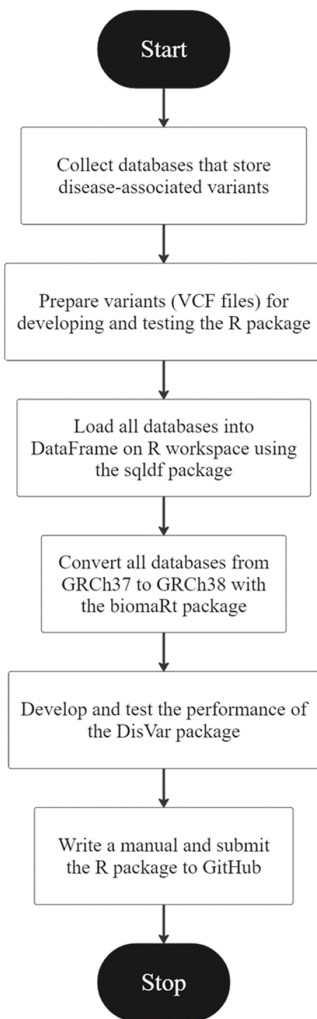

Figure 1   **An overview of the development of DisVar.**

RTX 3050 graphics card. All coding and testing of the R package was performed on this laptop using R version 4.3.0 and RStudio version 2023.03.1+524.

## DisVar development

An R package called "DisVar" was developed for identifying disease-associated variants from individual genetic information. The overview of the DisVar development workflow in Fig. 1 summarizes the following process. The GWAS Catalog, GWASdb, GRASP, GADCDC, and Johnson and O'Donnell databases were obtained. VCF files were prepared for coding the DisVar package. To prepare data for testing this R package, files were downloaded from the 1,000 Genomes Project, and the output format that would be obtained from processing the program was designed. Next, using the data.table package, a DataFrame on the R workspace was loaded with all the databases. Then, using the biomaRt package, all the databases were converted from GRCh37 to GRCh38 and saved as R Data (RDA) files. The

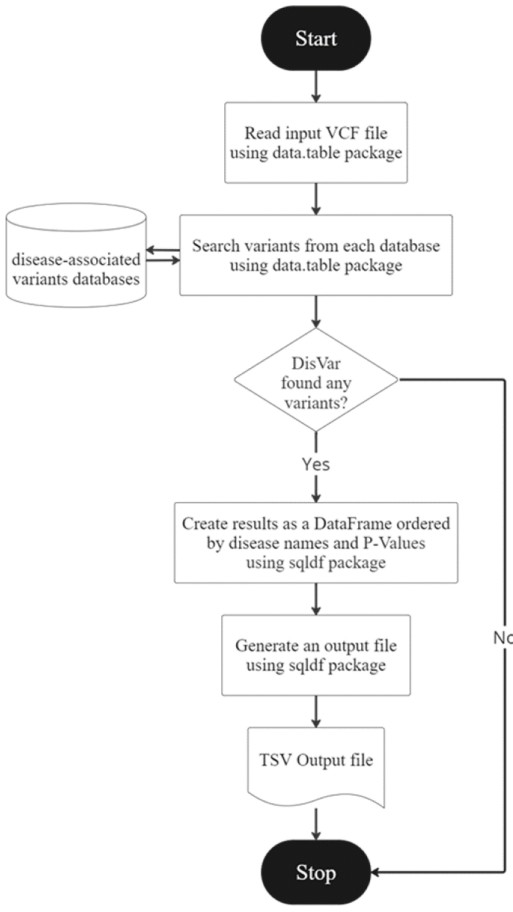

**Figure 2** **The DisVar flowchart.**

package was developed and modified to read large VCF files using the data.table package and disease-associated variants were searched using the data.table package. This involved implementing algorithms to search VCF files for the chromosomal position and location of SNP variants and to perform a database comparison to retrieve relevant information. In the first test of the package, the functionality of the algorithms was tested using a VCF file containing six to eight disease-associated variants. The package was then tested with a large dataset from the 1000 Genomes Project, comprising VCF files that contained from 215,054 to 11,346,899 variants, and the time elapsed in processing the data was measured using the system.time function. Finally, a manual for the package was written and submitted to GitHub.

DisVar is a tool for analyzing SNPs in genomic data. The DisVar flowchart is illustrated in Fig. 2. It begins by prompting the user to input a VCF file, which contains information about SNPs such as the chromosome number, position, and alleles. The function then reads this file and converts it into a data frame for further processing. Next, the function retrieves data from a pre-processed RDA file, which contains additional information about the SNPs such as the $P$-value (Confidence) and the GWAS trait, which is the trait associated with the

SNP in a GWAS. The data is stored in a separate data frame. The function then searches multiple databases for SNPs using the chromosome number and position as key criteria. Any SNPs that meet the criteria with a $P$-value less than $1.0 \times 10^{-7}$ are retrieved, along with the Rsid (a unique identifier for the SNP), the reference allele, the alternative allele, and the GWAS trait. If no matching SNPs are found in the databases, the user is informed, but if SNPs are found, a data frame is created and aligned according to the disease name, and diseases are sorted from the least to the greatest $p$-value. Finally, the function generates an output file in TSV (tab-separated values) format, which can be easily imported into other software for further analysis. The output file contains the disease, chromosome name, position, gene name, variant type, alleles of the variant in the sample, alleles of the variant in the database, $p$-value, database name, quality, filter, and info. GWAS trait for each SNP, arranged in a table according to the specified format. An example is shown in Table 2.

## RESULTS

The DisVar R package was subjected to a performance test using VCF files from the 1000 Genomes Project. The dataset consisted of 24 VCF files with different numbers of variants ranging from 215,054 to 11,346,899 sites. When testing the package, each VCF file was run separately. The total elapsed time was 63.58 min, and 298,227 hits of diseases-associated variants were returned across all 24 VCF files. The test results for each file are displayed in Table 3. The ClinVar VCF file was also examined. This file contains 2,120,558 total variant sites, and 20,657 hits associated with diseases were identified with an estimated elapsed time of 45.98 s.

A scatterplot and trendline were constructed of elapsed time in seconds *versus* the number of variants in millions using the Excel program (Fig. 3). The elapsed time increased as the variant size increased and there was a positive linear relationship between them. The linear regression model of trendline revealed a significant positive association between variant size and elapsed time (R-squared = 0.9). The regression equation was:

$$\textit{Elapsed time} = -39.78 + (3 \times 10^{-5} \times \textit{Number of variants})$$

The regression equation indicated that for every increase of one variant in input data, there was an average increase of 0.03 ms in elapsed time. These results suggest that our R package function performed well for small inputs of data but became slower for large inputs. Even so, the function can handle large VCF files. For example, a file with 5 million variants should take about 1.84 min to process.

## DISCUSSION

The DisVar R library was developed to overcome the limitations that currently exist in web applications. Compared to existing methods, this R package offers several advantages for identifying disease-associated variants. A comparison of program performance is shown in Table 4. First, DisVar is compatible with variants of the latest version of the reference genome and can simultaneously search for individual diseases in over 100 million sites. This

Chanasongkhram et al. (2023), *PeerJ*, DOI 10.7717/peerj.16086

**Table 2   Example of the output format.**

| Disease | Chr | Position | Gene | Variant_id | Variant_type | Allele Variant | Allele DB | *P*-value | DB | Qual | Filter | Info |
|---|---|---|---|---|---|---|---|---|---|---|---|---|
| Alzheimer's disease | 6 | 41161514 | TREM2 | rs75932628 | missense | C>T | C>T | 2.00E−12 | GWASdb | 7296.45 | PASS | AC=14;AF=0.00218613;AN=6404;BaseQRankSum=0.155; ClippingRankSum=0.481;DP=111734;... |
| | 8 | 95041772 | C8orf38 | rs7818382 | intron | C>T | C>T | 8.00E−08 | GWASdb | 1587730 | PASS | AC=3017;AF=0.471112;AN=6404;BaseQRankSum=0.223; ClippingRankSum=0;DP=106075;... |
| Breast cancer | 10 | 121577821 | FGFR2 | rs2981579 | intron | A>G | A>G | 2E-170 | GWASdb | 1491780 | PASS | AC=3227;AF=0.503904;AN=6404;BaseQRankSum=0.098; ClippingRankSum=-0.024;DP=95506;... |
| Nasopharyngeal carcinoma | 10 | 121577821 | FGFR2 | rs2981579 | intron | A>G | A>G | 2.00E−10 | GWASdb | 1491780 | PASS | AC=3227;AF=0.503904;AN=6404;BaseQRankSum=0.098; ClippingRankSum=-0.024;DP=95506;... |
**Table 3  The test results for each VCF file.**

| Chromosome | Variant | Elapsed time(s) | Hits found |
|---|---|---|---|
| 1 | 11,122,108 | 318.54 | 25,101 |
| 2 | 11,346,899 | 485.86 | 24,913 |
| 3 | 9,174,723 | 263.45 | 15,608 |
| 4 | 9,052,888 | 244.67 | 15,035 |
| 5 | 8,360,596 | 256.40 | 14,658 |
| 6 | 7,769,430 | 234.63 | 36,513 |
| 7 | 7,650,658 | 170.45 | 12,860 |
| 8 | 7,107,721 | 174.21 | 12,038 |
| 9 | 6,060,229 | 143.47 | 11,829 |
| 10 | 6,528,421 | 182.78 | 11,590 |
| 11 | 6,497,949 | 190.42 | 20,301 |
| 12 | 6,274,676 | 180.79 | 14,569 |
| 13 | 4,964,958 | 111.35 | 5,993 |
| 14 | 4,207,249 | 107.50 | 8,350 |
| 15 | 3,942,335 | 104.24 | 12,009 |
| 16 | 4,391,676 | 135.50 | 10,657 |
| 17 | 3,910,272 | 88.56 | 12,177 |
| 18 | 3,826,115 | 77.56 | 4,465 |
| 19 | 3,020,915 | 71.22 | 11,077 |
| 20 | 3,193,659 | 71.21 | 7,310 |
| 21 | 2,021,390 | 44.03 | 3,525 |
| 22 | 2,092,942 | 39.99 | 6,167 |
| X | 5,311,865 | 112.60 | 1,481 |
| Y | 215,054 | 5.54 | 1 |
| **Total** | **138,044,728** | **3,814.97** | **298,227** |

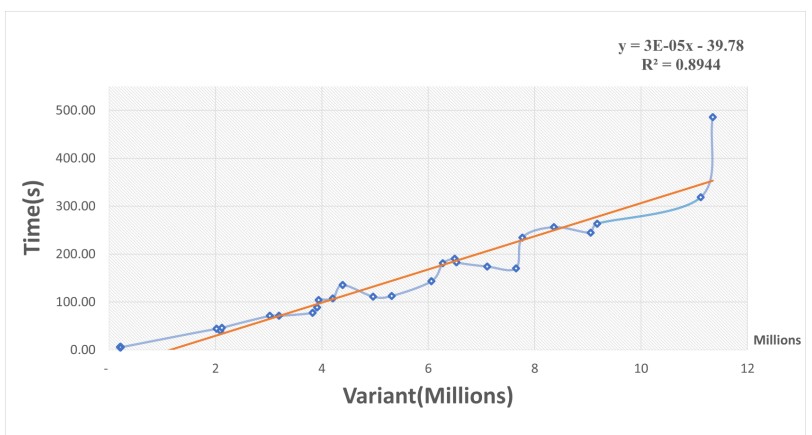

**Figure 3  Scatterplot of elapsed time (s) *versus* the number of variants (millions).**

**Table 4  A comparison of DisVar performance with other existing tools.**

| #Variants/Program name | DisVar | GWAS4D | MutationTaster2021 | VARAdb | GWAS Catalog |
|---|---|---|---|---|---|
| 1 | Yes | Yes | Yes | Yes | Yes |
| 100 | Yes | Yes | Yes | Yes | No |
| 40,000 | Yes | Yes | Yes | No | No |
| 11,346,899 | Yes | No | No | No | No |
| 138,044,728 (Multiple VCF files) | Yes | No | No | No | No |

is a significant improvement over web applications such as the GWAS Catalog, where vcf files are not supported. It can only search for single variants or a range of genome positions and does not support the variants of the latest reference genome. In addition, compared to VARAdb, which is similar to the GWAS Catalog, it can search for a genomic location or rsIDs, but VARAdb can search up to 100 variations. Furthermore, the results generated by DisVar are simple to understand and give users a clear understanding of the variants that are associated with a disease. In comparison, GWAS4D can simultaneously search approximately 40,000 variants but the results generated make it difficult to determine the disorder associated with a variant. Another limitation of GWAS4D is that it only shows the results for one variant at a time in each database and does not support VCF files as input. Therefore, users must check multiple databases and convert their VCF files to a compatible format before using GWAS4D. Similar to MutationTaster2021 is also able to support large files, but cannot support up to millions of positions, and neither of them can support VCF files derived from the latest reference genome. Finally, DisVar is able to process enormous amounts of data quickly and effectively, and handle multiple VCF files simultaneously. This is crucial in medical facilities or research settings where large amounts of data need to be promptly and precisely evaluated to identify genetic diseases. The functions of DisVar will be improved in the future, with a particular emphasis on its ability to analyze data from various sources and to incorporate new disease-associated variations. One potential advantage of this package is that it is an R package, which allows users to program and customize their analysis pipeline.

## CONCLUSIONS

An R package called DisVar was developed for identifying large-scale disease-associated genetic variants against the latest reference genome. This package simplifies the process of detecting genetic diseases based on an individual's genetic variants by enabling users to simultaneously search for individual diseases across over 100 million sites. Our study has shown that DisVar can process large data sets rapidly and effectively, and handle multiple VCF files simultaneously, providing a useful tool for medical facilities or research settings. It is a useful approach to identifying disease-associated variants from large-scale individual genomic information. Due to its ability to quickly and effectively analyze large volumes of data and its compatibility with the variants of the most recent reference genome, DisVar can help the identification and prevention of many diseases.

## ACKNOWLEDGEMENTS

We thank Mr. Thomas Coyne for language proofreading.

### Funding

Khunanon Chanasongkhram was supported by a Graduate Fellowship (Research Assistant), Faculty of Science, Prince of Songkla University. The funders had no role in study design, data collection and analysis, decision to publish, or preparation of the manuscript.

### Grant Disclosures

The following grant information was disclosed by the authors:
Graduate Fellowship, Faculty of Science, Prince of Songkla University.

### Competing Interests

The authors declare there are no competing interests.

### Author Contributions

- Khunanon Chanasongkhram conceived and designed the experiments, performed the experiments, analyzed the data, prepared figures and/or tables, authored or reviewed drafts of the article, and approved the final draft.
- Kasikrit Damkliang conceived and designed the experiments, authored or reviewed drafts of the article, and approved the final draft.
- Unitsa Sangket conceived and designed the experiments, performed the experiments, analyzed the data, prepared figures and/or tables, authored or reviewed drafts of the article, and approved the final draft.

### Data Availability

The DisVar package and its manual are available at GitHub and Zenodo:
- https://github.com/Khunanon-Chanasongkhram/DisVar.
- Khunanon Chanasongkhram, & unitsa-sangket. (2023). Khunanon-Chanasongkhram/DisVar: v1.1.4 (v1.1.4). Zenodo. https://doi.org/10.5281/zenodo.8280933.

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
