# Peer review of "DisVar: an R library for identifying variants associated with diseases using large-scale personal genetic information"

_PeerJ, doi:10.7717/peerj.16086_

## Round 0.1 · original submission · Major Revisions

The manuscript has been reviewed by two experts in the field. As you can see from their comments, both of them raise several points: reviewer 1 particularly emphasizes the necessity of the availability of the tool and the scripts used; I agree with the opinion, which is also required in the journal's policy. As the handling editor, I would like the authors to enlarge the Results section extensively. Currently, it is almost the same length as the results in the Abstract. Perhaps, the description of the tool in the Methods can be moved to Results. And I wonder if the authors could somehow add a comparison of its performance with that of other existing tools. Looking forward to the revised version.

Reviewer 1 ·

Basic reporting

This paper was clearly and unambiguously written in professional English. The structure of this paper conforms to PeerJ standards. Links to raw data were provided.

Experimental design

In this study, authors developed a novel R library called "DisVar" to identify disease-
associated genetic variants in large-scale individual genomic data, which is original primary research within Scope of the journal. The research topic is well defined, relevant, and meaningful.

Validity of the findings

The links to all underlying data have been provided; The methods are robust, statistically sound,and controlled.

In Abstract line 32 to 36, the authors claimed that their method "DisVar" detected 298,227 disease-associated variants hits in 1000 Genomes data in 3,689.21 s, and detected 20,657 hits in ClinVar data in 35.18 s. Since DisVar is run locally on an user's computer, the times it takes to detect hits from any data set could vary depending on the specification of users' computers. It's important for the authors to provide the specification of the computer they used to run the tests. Authors should provide the exact scripts they used to run the two tests they mentioned in supplemental.

Additional comments

It's highly suggested to the authors to make DisVar R package available to public through bioconductor or cran, which are the most popular R archive network.

Reviewer 2 ·

Basic reporting

Authors are well-described about the importance of the study.

Experimental design

No comment.

Validity of the findings

No comment.

Additional comments

This study is used bioinformatics tools for identifying variants associated with diseases. This will be very useful for medical biotechnology. However, I have suggested that as follows;
-The abstract should be added the conclusion and synthesized the importance of this tool in part of the abstract.
-For identifying variants, the variants (SNPs) should be assessed such as Synonymous SNP, Nonsynonymous SNP, as well.
-The author should add and more discuss about the advantage and disadvantage of bioinformatics tools due to too few references, resulting in weakness. This point is very important, please consider and improve.
- If you can tell where these variants (SNP) are located in the gene, this will reveal more associations with genetic diseases.
-Is there any criteria for selecting the quality of data variants in this research to be analyzed using this tool?

---

## Round 0.2 · accepted · Accept

Of the two assigned reviewers, one now recommends its acceptance. As for the minor, comments by the other reviewer, she/he has not agreed to check the revised manuscript but I confirmed that the authors addressed the raised points basically (I still believe that this service should become freely accessible). Thus, I would like to recommend its acceptance to the section editor.

Reviewer 2 ·

Basic reporting

No comment.

Experimental design

No comment

Validity of the findings

No comment

Additional comments

No comment